# Comparative Analysis of Chloroplast Genome of *Desmodium stryacifolium* with Closely Related Legume Genome from the Phaseoloid Clade

**DOI:** 10.3390/ijms24076072

**Published:** 2023-03-23

**Authors:** Le-Thi Yen, Muniba Kousar, Joonho Park

**Affiliations:** Department of Fine Chemistry, Seoul National University of Science and Technology, Seoul 01811, Republic of Korea

**Keywords:** *Desmodium styracifolium*, comparative chloroplast genome analysis, legumes, Phaseoloid, molecular marker

## Abstract

*Desmodium styracifolium* is a medicinal plant from the *Desmodieae* tribe, also known as *Grona styracifolia*. Its role in the treatment of urolithiasis, urinary infections, and cholelithiasis has previously been widely documented. The complete chloroplast genome sequence of *D. Styracifolium* is 149,155 bp in length with a GC content of 35.2%. It is composed of a large single copy (LSC) of 82,476 bp and a small single copy (SSC) of 18,439 bp, which are separated by a pair of inverted repeats (IR) of 24,120 bp each and has 128 genes. We performed a comparative analysis of the *D. styracifolium* cpDNA with the genome of previously investigated members of the *Sesamoidea* tribe and on the outgroup from its *Phaseolinae* sister tribe. The size of all seven cpDNAs ranged from 148,814 bp to 151,217 bp in length due to the contraction and expansion of the IR/SC boundaries. The gene orientation of the SSC region in *D. styracifolium* was inverted in comparison with the other six studied species. Furthermore, the sequence divergence of the IR regions was significantly lower than that of the LSC and the SSC, and five highly divergent regions, *trnL-UAA-trnT-UGU*, *psaJ-ycf4*, *psbE-petL*, *rpl36-rps8*, and *rpl32-trnL-UGA*, were identified that could be used as valuable molecular markers in future taxonomic studies and phylogenetic constructions.

## 1. Introduction

Chloroplasts are vital to plant organelles that function as metabolic centers. They possess their own genomes (cpDNA), which generally encode 110–130 genes and are essential for the photosynthesis pathways and the biosynthesis of nucleotides, fatty acids, starch, and pigments [1,2,3]. Plastid chromosomes are circularized, ranging from 120 to 160 kb [3], and contain a quadripartite architecture with two inverted repeat regions (IRs) that separate a large single-copy region (LSC) from a small single-copy region (SSC) [4,5]. The structural features and gene content of the plastome are highly conserved among land plants [6]. In addition, uniparental inheritance, low rates of substitution, and their small size, in comparison to the nuclear genome, facilitate their use as important models for evolutionary studies [7,8].

The Fabaceae or Leguminosae, commonly called the legume family, contain six subfamilies: *Caesalpinideae*, *Cercidoideae*, *Detarioideae*, *Dialioideae*, *Duoarquetioideae*, and *Papilionoideae* [9]. There has been morphological evidence and results from recent molecular phylogenetic studies suggesting that Fabaceae is a monophyletic family [10]. This family is composed of approximately 751 genera and over 19,500 species, and plays a crucial role in ecological habitat, boosting the economy, and providing a long-standing model for evolutionary studies [11].

The tribe, *Desmodieae*, is a large member of the Phaseoloid clade belonging to the *Papolionoideae* subfamily, which comprises the *Desmodium, Lespedeza, and Phyllodium* clades [12]. In several Asian regions, many species from this tribe are used as medicinal herbs because of their high levels of antioxidant and anti-inflammatory components [13,14,15]. Specifically, *Desmodium styracifolium*, from the tribe *Desmodieae*, is an important species in herbal medicine and has been employed to treat hepatitis and various types of urinary diseases such as renal stones, urinary tract infections, edema, and gallstone diseases [15,16,17]. While the whole plant has been used in traditional medicine, phytochemistry, and pharmacology, the complete chloroplast genome has not been previously sequenced or analyzed.

This investigation characterized the *Desmodium styracifolium* complete chloroplast genome and performed a comparative genome analysis with other *Desmodieae* species. We explored the taxonomic position of *D. styracifolium* as well as the genetic relationship between *Desmodium styracifolium* and other species in the same tribe, as it can serve as a reference for future studies of the *Desmodieae* tribe.

## 2. Results

### 2.1. Features of the D. styracrifolium Chloroplast Genome

The complete chloroplast genome of *D. styracifolium* was determined to be 149,155 bp in length and exhibits a four-region structure, with a pair of IR regions that are 24,120 bp each, separated by an LSC region and an SSC region with lengths of 82,476 bp and 18,439 bp, respectively. The complete genome harbors 128 genes, comprising 83 protein-coding genes, 37tRNA, and 8rRNA genes, in which LSC and SSC contain 82 genes and 12 genes, respectively, while 17 genes were duplicated in the IRs, shown in Figure 1. The protein-coding genes accounted for 55.3% (82,476 bp) of the total genome, whereas the remaining regions were composed of rRNA, tRNA, introns, and intergenic species.

In addition, we observed 19 intron-containing genes, in which 17 genes contained 1 intron (10 protein-coding genes and 7 tRNA genes) and 2 genes contained 2 introns (*ycf3* and *clnp*), *trnK-UUU* was found to have the largest intron, at 2626 bp, while the smallest intron found was in *trn4-UDG*, at 40 bp. The total GC content of the cpDNA was 35.2% and there were distinct differences between the 3 regions. The highest content exhibited in the IR regions with 42.1% was followed by the LSC, 32.8%, while the lowest content was observed in the SSC with 28.1%. There were 27,492 codons in the protein-coding genes, of which, leucine was the most common amino acid and was encoded by 11.3% of all codons.

### 2.2. Comparative Analysis with Other Chloroplast Genomes from the Desmodieae Tribe

#### 2.2.1. Gene Content

The *D. styracifolium* genome was compared to six *Desmodieae* species, namely *Desmodium heterocarpon* (NC_044113.1), *Hylodesmum podocarpum* (MG867568.1), *Lespedeza mariticma* (NC_044115.1), *Ohwia caudate* (NC_044105.1), and *Campylotropis macrocarpa* (NC_044100.1), alongside one outgroup, *Vigna radiata* (NC_013843.1). A detailed summary of their genomic features is shown in Table 1. The genome size ranged from 148,814 bp (*C. macrocarpa*) to 151,217 bp (*V. radiata*) in length. They exhibited typically quadripartite structures, consisting of the LSC (80,896–83,241 bp) and SSC (17,427–18,939 bp), separated by two IRs (23,720–26,474 bp).

In Table 1, a highly conserved structure was indicated in the *Desmodieae* complete chloroplast genome, regarding their gene content. This genome contained 128 genes, including 83 protein-coding, 37 tRNA genes, and 8 rRNA genes, while Vigna contained 127 genes with 1 tRNA (*trnG-GCC*) fewer than *Desmodieae*. There are 17 duplicated genes in *Desmodieae*, including 4 rRNA and 13 different genes (*rps12*, *rpl2*, *rpl23*, *trnI-CAU*, *ycf2*, *trnL-CAA*, *ndhB*, *rps7*, *trnV-GAC*, *trnI-GAU*, *trnA-UGC*, *trnR-ACG*, and *trnN-GUU*), while Vigna contained 1 more duplicated gene than Desmodium, *rps19*. Moreover, 19 intron-containing genes were found in the *Desmodieae* genomes. Two genes (*ycf3* and *clpP*) had two introns and the remaining seventeen had single introns, including nine protein-coding genes and eight tRNA genes. However, there were 20 intron-containing genes in *Vigna radiata*, owing to the presence of an intron in two *rpl2* genes and the loss of *trnG-UCC.*

#### 2.2.2. Gene/Intron Loss

This study found that *rpl22* and *infA* genes were lost in the seven chloroplast genomes due to their multiple transfers to the nucleus during evolution [18,19]. The seven chloroplast genomes were in accordance with this report. Moreover, the *ycf4* gene was non-functional in *D. heterocarpon* because of numerous internal stop codons and its lost initial start codon, although it is functional in the others. A similar observation was seen for the sequence of the *rps16* gene in *V. radiata*, which led to a loss of function. In addition, Vigna witnessed the *rpl33* gene as a non-functional gene due to the premature stop codon. As a result, *rpl22* and *infA* of *Faboideae*, the *ycf4* gene of *D. heterocarpon,* and the *rps16* and *rpl33* genes of *V. radiata,* were pseudogenes. Moreover, all Desmodieae species showed a loss of intron regions in the *rps12* and *rpl2* genes, whereas the *rpl2* was an intron-containing gene in *V. radiata*. and *rps12* was a trans-spliced gene in all species, with a 3′ end duplication in the IR regions and a 5′ end located in the LSC.

#### 2.2.3. Sequence Divergence Analysis

A structural characteristic comparison of *D. styracifolium* and six related species was performed using mVISTA. The results indicated that the *D. styracifolium* chloroplast genome was closer to *D. heterocarpon* and was most distinct from *Vigna radiata.*
Figure 2 showed in detail that the coding region was more conserved than the non-coding region and the IR regions had the lowest level of divergence, compared to the LSC and SSC regions. These results were in arrangement with the pattern demonstrated in other legumes—for example, *Lespedeza* [20], *Stryphnodendron* [21], and *Vigna* [22]. The elevated level of nucleotide divergence was observed in several genes, *matK/trnK*, *ycf4*, *rpl33*, *rpl16*, *ycf1*, and in the intron regions of *clpP* and *rps16*. Moreover, intergenic spacer regions were in remarkably divergent regions, such as between *trnH-GUG-psbA*, *trnK-rbcL*, *ndhJ-trnF-GAA*, *trnL UAA-trnT-UGU*, *petA-psbJ, psbE-petL*, *rps3-rps19*, *trnL-UAG-rpl32*, and *rpl32-ndhF*.

The nucleotide variability (Pi) values among six *Desmodieae* species were calculated using DnaSP software version 6.10.03. The mean Pi value was estimated to be 0.0472, ranging from 0 to 0.254, and revealed a noticeable divergence among the sequences. As expected, we observed a consistent result with the diversity reported for the comparison of the cpDNA sequences. Briefly, the results showed that the SSC contains the highest levels of nucleotide diversity, followed by the LSC, while the IRs were the most conserved regions (Figure 3).

The comparative analysis of the genome structure among the seven species revealed an SSC inversion in *D. styracifolium*. The gene orientation in the SSC region is shown in Figure 4A. In *D. styracifolium*, the gene *ycf1* was located at the IRb/SSC junction, then, followed by *rps15*, *ndhH*, *ndhA*, *ndhI*, *ndhG*, *ndhE*, *psaC*, *ndhD*, *ccsA*, *trnL*, *rpl32*, and *ycf1* at the IRb/SSC junction. The other six genomes had the completely same gene order as *D. styracifolium*, although in the reverse order.

To further confirm the orientation of the SSC regions in *D. styracifolium*, PCR amplification was employed. Two pairs of primers were designed to amplify the IR/SSC junction, shown in Figure 4B. At the IRa/SSC boundary, the GC content was exceptionally low for primer design; thus, half the length was in the IRa and the other half in the SSC, while the primer, P2R, was 180 bp away from the SSC boulder. Figure 4C illustrates that lanes 1 and 3 provided PCR products from the primer pairs of P1F/P1R and P2F/P2R, while no PCR product is shown in lane 2 from using the primers P1F/P2F.

#### 2.2.4. Repeat Sequence Analysis

The significant correlation between the number of repeat sequences with gene arrangement of the chloroplast genomes has previously been demonstrated [23]. Herein, a total of 325 tandem repeats were detected in the seven studied cpDNA genomes. *C. macrocarpa* contained the largest number of repeats (58), while *V. radiata* had the fewest (34). Among these repeats, 248 (76.3%) were in the intergenic spaces (IGS), followed by 59 (18.2%) in the coding regions, and the remaining 18 (5.5%) were found in the introns (Table 2). Repeat sequences in the IGS were abundant in *C. macrocarpa* (45), *D. heterocarpon* (42), *D. styracifolium* (41), and *L. maritima* (40), while the coding regions were highest in *O. caudate*. There were several genes in the coding regions that contained tandem repeats, namely, *ycf1*, *ycf2*, *ndhF*, *rps18*, *clpP*, *ndhA*, and *petB*. Tandem repeats were also detected in the intron regions of the two intron-containing genes (*rpl16* and *rpoC1*).

According to the quadripartite structure, most of these tandem repeats were distributed in the LSC region (64%), followed by 21.5% in the IR regions, and only 14.5% in the SSC region, which is represented in Figure 5A. The size of the repeat sequences mostly ranged from 31 bp to 50 bp, as seen in Figure 5B.

Simple sequence repeats (SSRs) were also analyzed using the MISA software tool. SSRs occur in all seven genomes, as seen in Table 3, and most were mono repeats composed of A/T repeats (337) and G/C repeats (4). A total of di- and trinucleotide repeats were rich in AT, with 82 AT/TA repeats and 3 AAT/TTA repeats, respectively. These findings show a strong AT level in SSRs, which had previously been observed in the Lespedeza tribe [20], as well as other legume species [21,24], suggesting a role for SSRs in the identification of these plant species.

#### 2.2.5. IR Expansion and Contraction

The length of the IRs was similar in *Desmodieae* species, ranging from 23,720 bp (*C. macrocarpa*) to 24,264 bp (*O. caudate*), whereas the outgroup *V. radiata* contained the longest IRs (26,474 bp), due to a duplicated gene (*rps19*). The boundary regions of the seven species are described in Figure 6.

It was noted that the IRa/LSC border in all six *Desmodieae* species was located downstream of the *trnH-GUG* gene and upstream of the *rpl2* gene by 1 bp (*D. heterocarpon*) to 34 bp (*D. styracifolium*), from the endpoint of the boundary to *trnH-GUG*. However, this junction in *V. radiata* was in the intergenic region of *rps19/rps3*. In addition, *ycf1* was present on the IRa/SSC junction in four species and the size of its fragments in IRa was largest in *V. radiata* (492 bp), followed by 463 bp in *O. caudate*, while that of *H. podocarpum*, and *D. heterocarpon* were 453 bp and 412 bp, respectively. On the other hand, the *ycf1* did not extend to the IRa region in *L. maritima* and *C. Macrocarpa* because they were 144 bp and 131 bp, respectively, away from the endpoint of the junction. However, *D. styracifolium* witnesses a presence of *ndhF* at the IRa/SSC boundary with no separation to the junction.

#### 2.2.6. Phylogenetic Relationship

In this study, the chloroplast genome from nine *Desmodieae* species and two outgroups were aligned using ETE3 [25] and the phylogeny reconstruction based on the RAxML analysis [26]. All eleven species produced two branches with dedicated support using bootstrap values of 100%. *V. radiata* and *M. macrocarpa* were in the *Phaseolinae* tribe, a sister group to *Desmodieae*. The remaining nine *Desmodieae* species were divided into three clades. *O. caudata* belongs to the genus *Phyllodium*. The second clade consisted of four taxa grouped to the genus *Desmodium*, including *H. podocarpum*, *D. heterocarpon*, *D. styracifolium. C. macrocarpa*, *K. striata*, *L. maritima*, *L. davurica*, and *L. floribunda*, which were grouped into the third clade, *Lespedeza* (Figure 7).

## 3. Discussion

The study was conducted as a comparative analysis of the complete chloroplast genome of *D. styracifolium*, with five species of the *Desmodieae* tribe and an outgroup. The results indicated that the chloroplast characteristics among the *Desmodieae* species were highly conserved, which was similar to a previous study [27], where similar features among the members of *Desmodieae* were also mentioned. However, a distinct feature was also identified in the Desmodium clade, which was the pseudogene, ycf4, in *D. heterocarpon*, however, it remained a functional gene in *D. styracifolium* and *H. podocarpum*. The ycf4 gene encoded the Ycf4 protein, which is involved in the biogenesis and assembly of the photosystem I (PS I) complex, although it was a nonessential factor for PSI activities in higher plants [28,29]. Furthermore, the evolution of the *ycf4* gene may be varied in legumes, as it was considered to be a localized hypermutation in several legumes species, resulting in gene losses in the chloroplast genome [30].

The gene content of the *D. styracifolium*, when compared with the other six species, found that *O. caundata* have the largest coding size, followed by *D. styracifolium*, while *L. maritima* have the smallest. The remarkable reasons for the chloroplast genome size variations are the intergenic region variations, the shrinkage and expansion of the IR regions, and the gene/intron loss, which were consistent with a previous report [31]. Moreover, the GC content ranged from 34.9% in *C. macrocarpa* to 35.4% in *V. radiata*. The three Desmodium species had 35.2% GC content. The IR regions contained the highest percentage of GC because the four pairs of rRNA genes were in those regions. It was previously demonstrated that the *rpl22* and *infA* genes were lost in all legumes due to its multiple transfers to the nucleus during evolution [18,19]. The seven chloroplast genomes showed some resemblance to this report. The crucial role of RNA editing in the translation process was observed in all mentioned species. In *ndhD* transcripts, the C-to-U editing was shown, resulting in the conversion of the start codon from ACG to AUG. The chloroplast RNA editing seemed to function as a mechanism to generate variations at the RNA and DNA levels [32].

Moreover, according to the results of the SSC inversion in *D. styracifolium*, which resulted in high divergences in the SSC region [Figure 3], suggesting that the sequence arrangements occurred in the chloroplast genome during evolution, even though conservation of the gene order in the chloroplast genome had previously been reported [6,33]. Furthermore, the number and size of the largest repeat sequences were significantly associated with the degree of genome recombination. The size of the repeat sequence had been observed in the range of 31–50 bp, which was in accordance with other legume species [34]. Both the smallest repeat (23 bp) and the longest repeat (175 bp) were found in *D. heterocarpon*. Repeats larger than 70 bp were absent in *O. caudate* and *L. maritima*.

SSRs consisted of tandemly repeated mono-, di-, tri-, or tetranucleotide motifs [35]. SSRs were useful genetic markers that could play a vital role in assessing chloroplast variations and for phylogenetic relationship studies [36]. The shift in the boundary between IR regions and single-copy regions was thought to cause size variations in the chloroplast genomes. At the border of the IRb/SSC junction, the distance from the *ndhF* end to the boulder was 24 bp in *C. macrocarpa*, 16 bp in *L. maritima*, and 5 bp in *D. heterocarpon*, while *ycf1* extended to the IRb by 411 bp in *D. styracifolium.* The sizes of the *ndhF* gene fragments in the IRb were 23 bp in *V. radiata*, and 13 bp more than in *H. podocarpum* and *O. caudate*. Furthermore, *rps19* crossed the LSC/IRb boundary in most studied species, except for *V. radiata*, in which *rps19* was located 592 bp upstream of the boulder. The IRb region differently expanded to *rps19* among the *Desmodieae* species, by 16 bp in *O. caudate* to 56 bp in *D. heterocarpon*. The expansion of IRs in *V. radiata* caused the duplication of *rps19*, the smaller size of its SSC, and the larger size of the IR regions, thus, the complete genome size was largest in these seven studied species.

DNA barcoding was the cost-effective and highly accurate biotechnology method for species identification by analyzing 400–800 bp long specific DNA regions, which could be mitochondrial, plastidial, or nuclear original genes. There were a number of chloroplast-derived DNA barcodes that had been used for phylogenetic analysis, such as *rbcL*, *matK*, *rpoB*, *psbA-trnH*, and *atpF-atpH.* In the tribe, Desmodiae, a short region of internal transcribed spacer (ITS) and a combination of *rbcL* + *matK* + ITS had been used as markers for the species-level identification, although these barcodes still had the limitation relating to the availability of the sequences and the rate of identification success. In this study, the Pi value was calculated to identify the highly divergent regions, *trnL-UAA-trnT-UGU*, *psaJ-ycf4*, *psbE-petL*, *rpl36-rps8*, and *rpl32-trnL-UGA* that could be utilized as DNA barcodes for investigating the phylogenetic relationship of Desmodiae.

In conclusion, the plastome size, GC content, gene orientation, gene content, tandem repeats, and microsatellites were highly conserved in the studied species, except for the SSC inversions in *D. styracifolium*. However, the IR/SC boundaries and sequence divergence showed differences among these species. The comparison of the chloroplast genomes indicated that the coding regions were more conserved than the non-coding regions, and the divergence level of the IRs was lower than that of the LSC and SSC, which is consistent with the results previously found for other legume species. Five variable regions (*trnL-UAA-trnT-UGU*, *psaJ-ycf4*, *psbE-petL*, *rpl36-rps8*, and *rpl32-trnL-UGA*) were identified, which may be useful as potential markers for further phylogenetic analysis and evolutionary studies. Overall, this research offers a useful method for choosing chloroplast markers, examining obscure phylogenetic relationships, and avoiding taxonomic mistakes brought on by significant sequence variations.

## 4. Materials and Methods

### 4.1. Plant Materials and DNA Extractions

Young and healthy leaves of *Desmodium styracifolium* were collected from the National Institute of Biological Resources, Incheon, South Korea (NIBRGR000001122251) and stored at −80 °C until use. The total genomic DNA was immediately extracted by using samples of 1 g of Desmodium leaves in the CTAB method, as previously described by [37]. The DNA concentration and quality were assessed using spectrophotometry and 1% (*w*/*v*) agarose gel electrophoresis.

### 4.2. DNA Sequencing, and Genome Assembly and Annotation

The extracted total genome DNA of D. styracifolium was sequenced for the complete genome using the PacBio RS II system (Pacific Bioscience Inc., Menlo park, CA, USA). The adapter sequences from the raw data were removed to obtain high-quality subreads. Filtered subreads were first mapped to the D. heterokaryon (NC_044113.1) chloroplast genome from the NCBI database using the BWA Aligner. The matched subreads were assembled for the chloroplast genome with CANU version 1.8 followed by checking the overlapped regions using nucmer and nummerplot, and they were, then, annotated with the GESqeq Annotation tool. The circular genome map was drawn with OGDraw software, version number 1.3.1 [38]. The complete DNA sequence was deposited in the GenBank database under the accession number: MN913536.

### 4.3. Sequence Analysis

Genomes of six other species were obtained from the GenBank with the following accession numbers: *Desmodium heterocarpon* (NC_044113), *Hylodesmum podocarpum* (MG867568), *Lespedeza maritima* (NC_044115), *ohwia caudate* (NC_044105) [22], and *Vigna radiata* (NC_013843) [27]. The complete cpDNA of *D. Styracifolium* was compared to these species. The online comparison tool mVISTA [39] in the shuttle-LAGAN mode was chosen to perform sequence alignment analysis. The *D. styracifolium* and V. radiata plastomes were used as the reference and out group, respectively. To evaluate the nucleotide variability (Pi) among the six plastomes of the *Desmodieae* species, the sequences were initially aligned using CLUSTALX 1.81 [40]. Then, a sliding window analysis was performed to calculate the nucleotide diversity using DnaSP version 6.10.03, with a step size of 200 bp and window length of 600 bp [41].

Simple sequence repeats (SSRs) or microsatellites were determined using the online MISA (Microsatellite identification tool (https://webblast.ipk-gatersleben.de/misa/ accessed on 16 January 2020) [42]. The minimum numbers of repetitions of mono-, di-, tri-, tetra-, penta-, and hexanucleotide were 10, 6, 5, 5, 5, and 5, respectively. Tandem Repeats Finder (TRF) version 4.09 was used to detect the tandem repeats.

### 4.4. PCR Amplification

To validate the assembly at the IRs and SSC junction regions in *D. styracifolium*, we designed four primers, described in Table 4, to amplify the IR/SSC junctions. The PCR amplification reactions were performed using the BioFACTTM 2× Real-time PCR Master Mix (including SFCgreen I), with 15 min incubation at 95 °C, followed by 30 cycles of 30 s denaturation at 95 °C, 30 s annealing at 59 °C, and 50 s extension at 72 °C. Each 20 µL PCR reaction system included 10 µL of the Master Mix, 2 µM Primers, and 100 µg DNA. PCR products were separated by running them on 1% (*w*/*v*) agarose gel electrophoresis. A PCR 100 bp (Sigma-Aldrich) was used as the molecular weight marker.

## Figures and Tables

**Figure 1 ijms-24-06072-f001:**
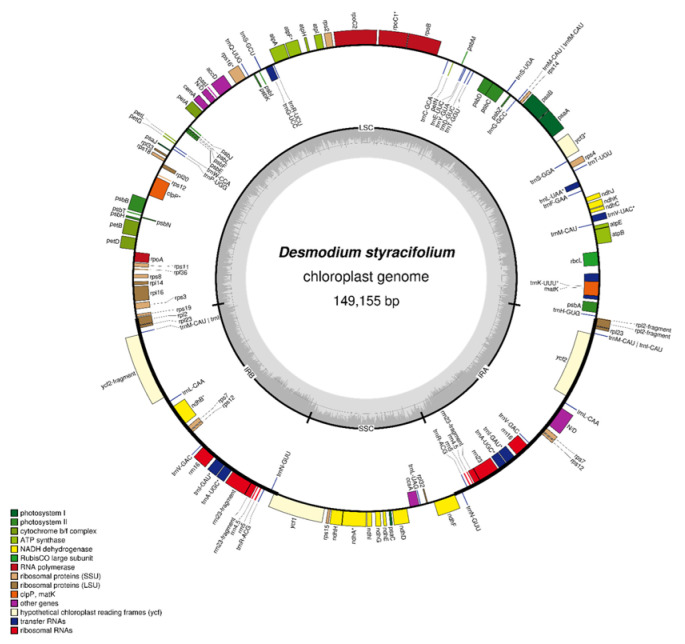
Gene map of *Desmodium styracifolium*. The genes outside the circle are transcribed counterclockwise, while those inside is transcribed clockwise. Genes are color based on the identified functional groups.

**Figure 2 ijms-24-06072-f002:**
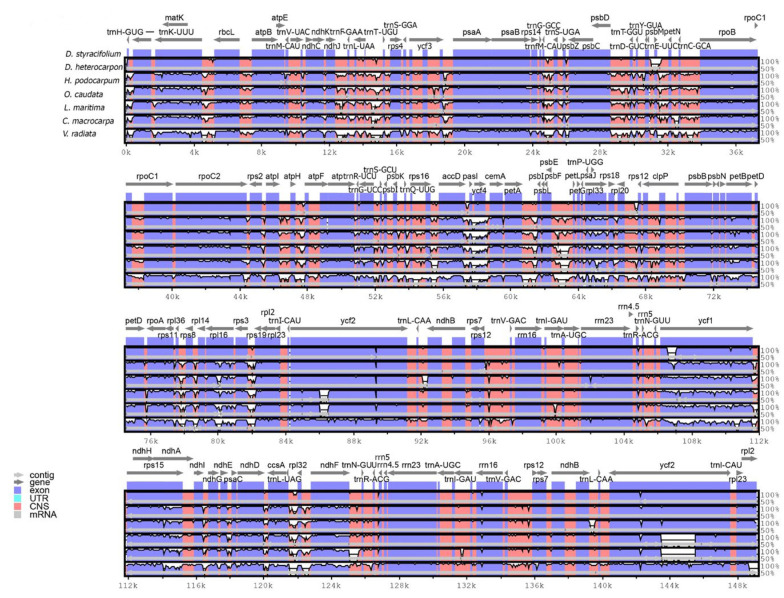
Sequence identify plot for seven Phaseoloid species using mVISTA with *Desmodium styracifolium* as a reference. Gray arrows represent gene orientation and position. The Y-axis indicates the percentage identity between 50 and 100%. Red and blue bars indicate non-coding sequences (CNSs) and exons, respectively.

**Figure 3 ijms-24-06072-f003:**
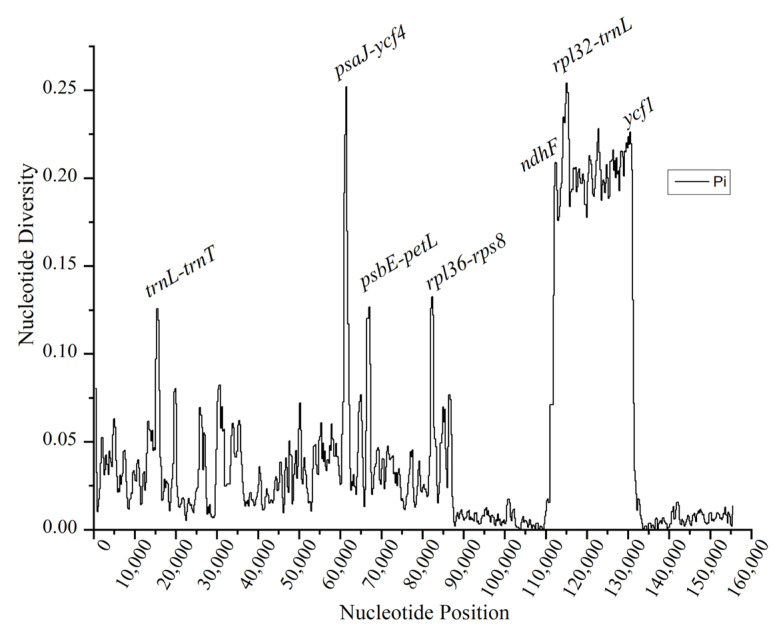
Nucleotide variability (Pi) values of the six *Desmodieae* chloroplast genomes using a sliding window analysis.

**Figure 4 ijms-24-06072-f004:**
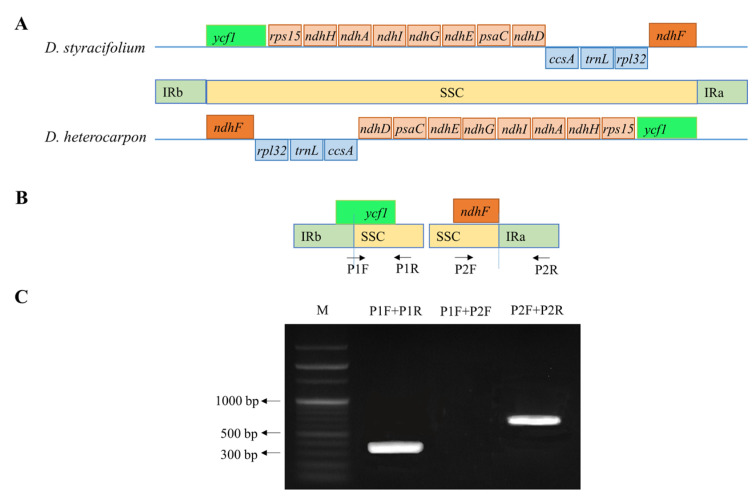
Analysis of the SSC inversion in *Desmodium styracifolium.* (**A**) Comparison of the gene order in the SSC region of *D. styracifolium* and *D. heterocarpon*. (**B**). Primer design for the PCR to amplify the IR/SSC junction regions in *D. styracifolium*. Two pairs of primers, primer 1 (P1F/P1R) and primer 2 (P2F/P2R) are designed to amplify the IRb/SSC junction. Primer P1F is located at the IRb/SSC boundary, while P1R and P2R are in *ycf1* and *ndhF,* respectively, which are in the SSC region. (**C**) PCR amplification of the IR/SSC junctions in *D. styracifolium*. M: PCR 100 bp Low Ladder.

**Figure 5 ijms-24-06072-f005:**
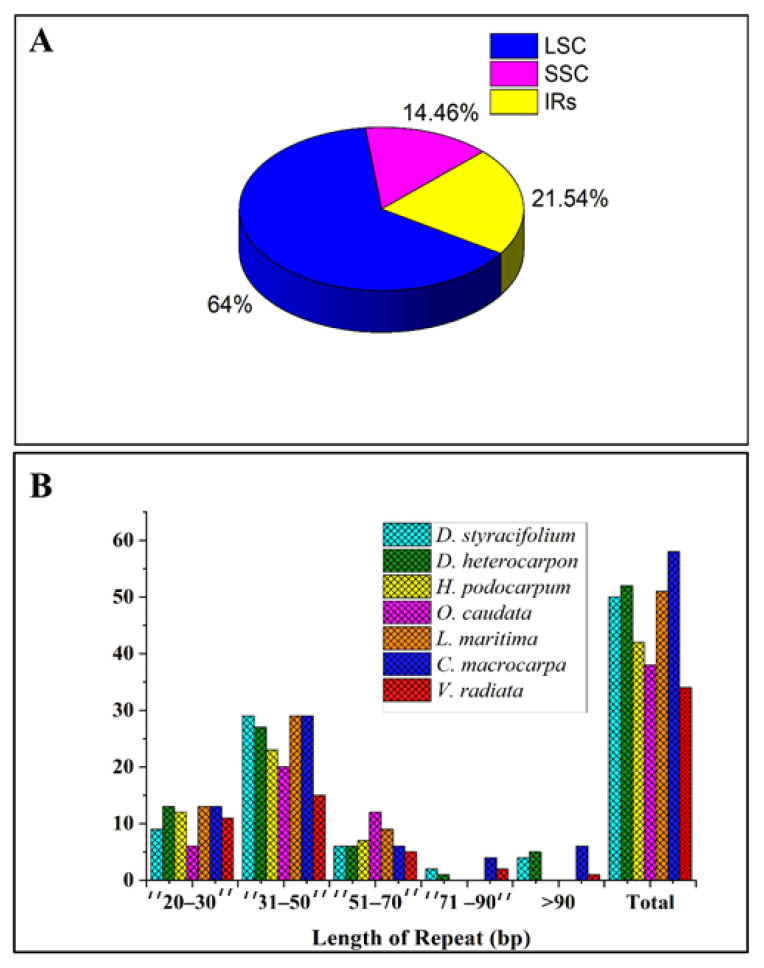
Tandem repeat analysis among seven studied species. (**A**) Percentage distribution of all tandem repeats based on the quadripartite structure. (**B**) Frequency of tandem repeats by length.

**Figure 6 ijms-24-06072-f006:**
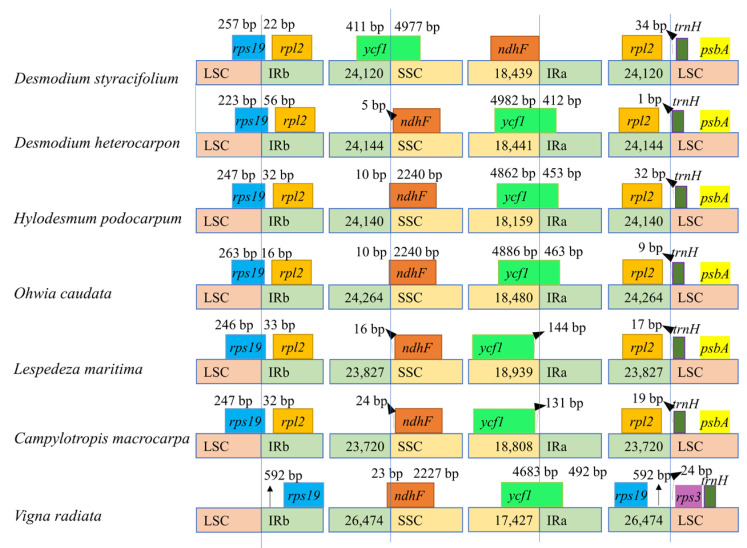
Comparison of the IR/SC junctions among seven chloroplast genomes.

**Figure 7 ijms-24-06072-f007:**
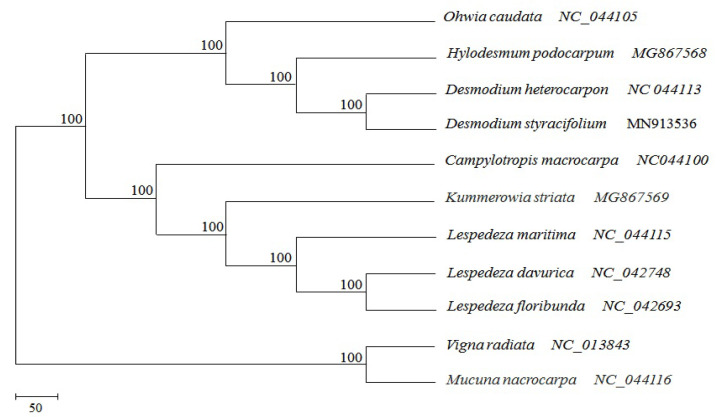
Maximum-likelihood phylogenetic tree reconstruction based on the complete chloroplast genome of eleven species.

**Table 1 ijms-24-06072-t001:** Summary of the chloroplast genome features of six *Desmodieae* and an outgroup (*Phaselous vulgaris*).

Features	*D. styracifolium*	*D. heterocarpon*	*H. podocarpum*	*O. caudata*	*L. maritima*	*C. macrocarpa*	*Vigna radiata*
Genome size	149,155	149,696	149,564	150,249	149,022	148,814	151,271
LSC Length	82,476	82,967	83,125	83,241	82,429	82,566	80,896
SSC Length	18,439	18,441	18,159	18,480	18,939	18,808	17,427
IR length	24,120	24,144	24,140	24,264	23,827	23,720	26,474
Coding Size	78,033	77,427	77,925	78,045	77,265	77,925	77,469
GC content	35.2	35.2	35.2	35.1	35.0	34.9	35.4
Total genes	128	128	128	128	128	128	127
Protein-coding genes	83	83	83	83	83	83	83
Duplicated genes	17	17	17	17	17	17	19
tRNA genes	37	37	37	37	37	37	36
rRNA genes	8	8	8	8	8	8	8
Genes-with introns	19	19	19	19	19	19	18
Pseudogenes		*ycf4*					*rpl33*, *rps16*

**Table 2 ijms-24-06072-t002:** Tandem repeat distribution among six Desmodieae species and an outgroup (Vigna).

	Intergenic Space	Coding Region	Region of Introns
*Desmodium styracifolium*	41	8	1
*Desmodium heterocarpon*	42	8	2
*Hylodesmum podocarpum*	33	7	2
*Ohwia caudate*	23	11	4
*Lespedeza maritima*	40	8	3
*Campylotropis macrocarpa*	45	9	4
*Vigna radiata*	24	8	2
Total	248	59	18

**Table 3 ijms-24-06072-t003:** Type and number of SSRs in the chloroplast genome.

SSR Type	Repeat Unit	*D. styracifolium*	*D. heterocarpon*	*H. podocarpum*	*O. caudata*	*L. maritima*	*C. macrocarpa*	*V. radiata*	Total
Mono	A/T	49	50	56	55	44	45	38	337
	C/G	0	0	2	0	2	0	0	4
Di	AT/AT	8	10	32	8	7	12	5	82
Tri	AAT/ATT	0	0	0	0	0	2	1	3

**Table 4 ijms-24-06072-t004:** PCR primers used in PCR amplification.

Primer	Sequence
PIF	TCTTATTTTTCAGTTATTAAATCCCCT
PIR	GGGGGCGGTATTCTACCTA
P2F	ACGCAACTTTTTCAGTGATTCT
P2R	GCGCCTTTGCATCTAGCATT

## Data Availability

All other relevant data are available from the corresponding author upon reasonable request.

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
