# Peer review of "Comparative Analysis of Chloroplast Genome of *Desmodium stryacifolium* with Closely Related Legume Genome from the Phaseoloid Clade"

_ijms, 2023, doi:10.3390/ijms24076072_

Round 1

Reviewer 1 Report

The work by Park et al. (Comparative Analysis of Chloroplast Genome of Desmodium stryacifolium with Closely Related Legume Genome from the Phaseoloid Clade)) reports the characterization of the Desmodium styracifolium complete chloroplast genome and performs a comparative genome analysis with other Desmodieae species. For this, the complete chloroplast genome of Desmodium stryacifolium was extracted and sequence analyses were performed to find out the taxonomic position and phylogenetic relationships of Desmodium stryacifolium with other  Desmodium species.

Although the manuscript is well-written and easy to follow, the corrections listed below should be made before publication.

1.     In the Abstract, please write the full version of the abbreviations.

2.     Why did you use the maximum likelihood for phylogenetic analysis?

3.     I noticed some grammatical errors in the entire manuscript. Please correct them.

4.     In the discussion part, please complete the second sentence.

5.     Discussion should be extended.

6.     In the discussion part or separately, it is better to add conclusion sentences.

Author Response

Dear Reviewer 1,

Thank you for giving us the opportunity to submit a revised draft of our manuscript titled ‘’Comparative Analysis of Chloroplast Genome of Desmodium Styracifolium with closely related legume Genome from the Phaseoloid Clade’’ to ‘’International Journal of Molecular Sciences’’. We appreciate the time and effort that you and reviewers have dedicated to providing your valuable feedback on our manuscript. We are very grateful to the reviewers for their insightful comments on my paper. We have been able to incorporate the changes to reflect most of the suggestions provided by the reviewers and have highlighted the changes within the manuscript. We hope that all the changes and explanations satisfy the reviewers and the editor. Thank you

Here is a point-by point response to the reviewer’s comments and concerns.

Comments from Reviewer 1.

  • Comment 1: In the Abstract, please write the full version of the abbreviations.

Response: Thank you for pointing this out. I agree with this comment. Therefore, we updated the manuscript by mentioning the full form of our abbreviation, for example from line no 12 - 13, we changed the manuscript as, it is composed of a large single copy (LSC) 82,476bp and small single copy (SSC) 18,439bp, separated by a pair of inverted repeats (IR) 24,120bp each, and has 128 genes.

  • Comment 2: Why did you use the maximum likelihood for phylogenetic analysis?

Response: we used the maximum likelihood for phylogenetic analysis, because this approach guarantees the consistency of the maximum likelihood (ML) estimate of the branch lengths of a phylogenetic tree, given the correct tree topology and nucleotide substitution model. Moreover, it provides the probability of a relationship that is the most likely to occur among the individuals. It used different parameters of statistics, which makes it even more acceptable.

  • Comment 3: I noticed some grammatical errors in the entire manuscript. Please correct them.

Response: We have carefully checked the manuscript to eradicate any grammatical errors.

  • Comment 4: In the discussion part, please complete the second sentence.

Response: Agree, we revised the manuscript to emphasize this point. We changed the manuscript as, the results indicate that the chloroplast characteristics among the Desmodieae species were highly conserved, which is similar with the previous study [27], in which they mentioned about the similar features among the members of Desmodieae. Line no 229-232 in a discussion section.

Comment 5: Discussion should be extended.

Response: the authors appreciate the reviewer for this comment, we have extended the discussion section accordingly, and highlighted all the changes in the manuscript, page 10.

Comment 6: In the discussion part or separately, it is better to add conclusion sentences.

Response: we agree this, and we have incorporated your suggestion by adding a conclusion section in our manuscript in page 10. 

"In conclusion, the plastome size, GC content, Gene orientation, gene content, tandem repeats, and microsatellites were highly conserved in the studied species, except for the SSC inversions in D.styracifolium. However, the IR/SC boundaries and sequence divergence, showed differences among these species. The comparison of the chloroplast genomes indicated that the coding regions were more conserved than the non-coding regions, and the divergence level of the IRs was lower than that of the LSC and SSC; consistent with the results previously found for other legume species. Five variable regions (trnL-UAA-trnT-UGU, psaJ-ycf4, psbE-petL, rpl36-rps8, and rpl32-trnL-UGA) were identified, which may be useful as potential markers for further phylogenetic analysis and evolutionary studies. Overall, this research offers a useful method for choosing chloroplast markers, examining obscure phylogenetic relationships, and avoiding taxonomic mistakes brought on by significant sequence variations."

Reviewer 2 Report

I checked your manuscript and described comments below.

Desmodium styracifolium (Grona styracifolia) is important as a medicinal herb. This medicinal herb has diuretic and calculus excretion effects.

I think you should fix the following points.

1.       Desmodium styracifolium should be called Desmodium styracifolium (Grona styracifolia).

2.       NC_044113, NC_044115, NC_044105, NC_044100, NC_013843, these Refseq IDs should be NC_044113.1, NC_044115.1, NC_044105.1, NC_044100.1, NC_013843.1.

3.       M867568 is no longer available. Hylodesmum podocarpum (M867568) should be changed to Hylodesmum podocarpum (MG867568.1)

4.       MISA should be MISA (MICroSAtellite identification tool, https://webblast.ipk-gatersleben.de/misa/) and add the following references.

Beier S, Thiel T, Münch T, Scholz U, Mascher M (2017) MISA-web: a web server for microsatellite prediction. Bioinformatics 33 2583–2585. dx.doi.org/10.1093/bioinformatics/btx198.

5.       Line 305 has Careless Miss. accession numvers -> accession numbers

I don't think this paper has any major mistakes or grammatical problems.

Author Response

Dear reviewer 2,

Thankyou for giving us the opportunity to submit a revised draft of our manuscript titled ‘’Comparative Analysis of Chloroplast Genome of Desmodium Styracifolium with closely related legume Genome from the Phaseoloid Clade’’ to ‘’International Journal of Molecular Sciences’’. We appreciate the time and effort that you and reviewers have dedicated to providing your valuable feedback on our manuscript. We are very grateful to the reviewers for their insightful comments on my paper. We have been able to incorporate the changes to reflect most of the suggestions provided by the reviewers and have highlighted the changes within the manuscript. We hope that all the changes and explanations satisfy the reviewers and the editor. Thank you

Here is a point-by point response to the reviewer’s comments and concerns.

Comments from Reviewer 2.

Comment 1: Desmodium styracifolium should be called Desmodium styracifolium (Grona styracifolia).

Response: Agree, we changed the abstract portion of our manuscript as ‘’Desmodium styracifolium is a medicinal plant from the Desmodieae tribe also known as Grona styracifolia. Line number 9-10

Comment 2: NC_044113, NC_044115, NC_044105, NC_044100, NC_013843, these Refseq IDs should be NC_044113.1, NC_044115.1, NC_044105.1, NC_044100.1, NC_013843.1.

Response: Thank you for pointing these errors. I agree and change my manuscript according to your suggestion as NC_044113.1, NC_044115.1, NC_044105.1, NC_044100.1, NC_013843.1. line number 85-87.

Comment 3: M867568 is no longer available. Hylodesmum podocarpum (M867568) should be changed to Hylodesmum podocarpum (MG867568.1)

Response: I changed the accession number of Hylodesmum podocarpum from M867568 to (MG867568.1). Line no 86.

Comment 4: MISA should be MISA (MICroSAtellite identification tool, https://webblast.ipk-gatersleben.de/misa/) and add the following references.

Response: According to your suggestion the manuscript has been changed to MISA (Microsatellite identification tool (https://webblast.ipk-gatersleben.de/misa/) and references was added. Reference Number 42

Comment 5:  Line 305 has Careless Miss. accession numvers -> accession numbers.

Response: Thank you for this correction. We incorporated your suggestion throughout your manuscript.